# The Application of 3base™ Technology to Diagnose Eight of the Most Clinically Important Gastrointestinal Protozoan Infections

**DOI:** 10.3390/ijms241713387

**Published:** 2023-08-29

**Authors:** Mahdis Aghazadeh, Meghan Jones, Suneth Perera, Jiny Nair, Litty Tan, Brett Clark, Angela Curtis, Jackson Jones, Justin Ellem, Tom Olma, Damien Stark, John Melki, Neralie Coulston, Rohan Baker, Douglas Millar

**Affiliations:** 1Genetic Signatures, 7 Eliza Street, Newtown, NSW 2042, Australia; mahdis.aghazadeh@geneticsignatures.com (M.A.); meghan.jones@geneticsignatures.com (M.J.); suneth.perera@geneticsignatures.com (S.P.); jiny.nair@geneticsignatures.com (J.N.); brett.clark@geneticsignatures.com (B.C.); angela.curtis@geneticsignatures.com (A.C.); jackson.jones@geneticsignatures.com (J.J.); john.melki@geneticsignatures.com (J.M.); neralie.coulston@geneticsignatures.com (N.C.); rohan.baker@geneticsignatures.com (R.B.); 2Microbiology Department, Westmead Hospital, Westmead, NSW 2145, Australia; justin.ellem@health.nsw.gov.au (J.E.); tom.olma@health.nsw.gov.au (T.O.); 3St. Vincent’s Pathology, Level 6, Xavier Building, 390 Victoria Street, Darlinghurst, NSW 2010, Australia; damien.stark@svha.org.au

**Keywords:** gastrointestinal parasites, RT-PCR, genomic simplification

## Abstract

Globally, over 3.5 billion people are infected with intestinal parasites each year, resulting in over 200,000 deaths. Three of the most common protozoan pathogens that affect the gastrointestinal tract of humans are *Cryptosporidium* spp., *Giardia intestinalis*, and *Entamoeba histolytica*. Other protozoan agents that have been implicated in gastroenteritis in humans include *Cyclospora cayetanensis*, *Dientamoeba fragilis*, *Blastocystis hominis*, and the microsporidia *Enterocytozoon bieneusi* and *Encephalitozoon intestinalis*. Genetic Signatures previously developed a 3base™ multiplexed Real-Time PCR (mRT-PCR) enteric protozoan kit (EP001) for the detection of *Giardia intestinalis*/*lamblia*/*duodenalis*, *Cryptosporidium* spp., *E. histolytica*, *D*. *fragilis*, and *B. hominis*. We now describe improvements to this kit to produce a more comprehensive assay, including *C. cayetanensis*, *E. bieneusi*, and *E. intestinalis*, termed EP005. The clinical performance of EP005 was assessed using a set of 380 clinical samples against a commercially available PCR test and other in-house nucleic acid amplification tests where commercial tests were not available. All methods provided at least 90% agreement. EP005 had no cross-reactivity against 82 organisms commonly found in the gut. The EP005 method streamlines the detection of gastrointestinal parasites and addresses the many challenges of traditional microscopic detection, resulting in cost savings and significant improvements in patient care.

## 1. Introduction

The history of human parasitic infections dates to the Greek, Egyptian, and Roman Empires [1]. Infections with parasites, in particular protozoan agents, are a major health concern worldwide and cause over 200,000 deaths annually [2,3]. Intestinal protozoa are one of the most widespread infections in resource-limited countries and children, especially those under 5 years old [4], who are the most vulnerable population, as they tend to be more active in areas of high prevalence and not always adhere to strict hygienic practices [5]. Infections with *Giardia intestinalis*/*lamblia*/*duodenalis* (synonymous and referred to as *G. intestinalis* henceforth) and *Cryptosporidium* spp. are mainly transmitted to the population via water or food contaminated with cysts, and infection is typically characterized by acute diarrhea [6]. It has been estimated that Giardia infects more than 200 million individuals every year globally, and studies have suggested that chronic infection with Giardia in children can result in significant long-term growth retardation [7].

*Entamoeba histolytica* is the causative agent of amoebiasis and is responsible for dysentery and liver abscesses, with invasive disease developing in approximately 10% of infected individuals. The disease is endemic in many countries, including Central and South America, Africa, and Asia [8]. *E. histolytica* is transmitted to the population via contaminated water, foods, and through sexual contact [9].

*Dientamoeba fragilis* and *Blastocystis hominis* have also been implicated as intestinal protozoan capable of causing disease in humans. The prevalence of *D. fragilis* has been found to be between 0.4% and 71% depending on the population studied and the diagnostic methods used; in general, the prevalence is higher in developed countries [10]. Clinical symptoms range from asymptomatic to abdominal pain, weight loss, diarrhea, and altered bowel movements consistent with gastroenteritis [10]. It has been postulated that *D. fragilis* could be a heterogeneous species containing variants possessing different pathologies [11]. *Blastocystis hominis* is a controversial human pathogen, and the prevalence has been shown to be 0.5–24% in industrialized countries and 30–76% in resource-limited nations [12,13]. There have been many reports suggesting that *B. hominis* causes disease, and equally a number of reports that contradict this claim [14]. Nevertheless, *B. hominis* has been included in the Water Sanitation and Health programs of the World Health Organization, indicating its pathogenic potential [15]. Furthermore, recently, it was postulated that *B. hominis* subtypes ST1 and ST3 and *Cryptosporidium* spp. might be associated with colorectal cancer in a systematic review of the literature [16].

*Cyclospora cayetanensis* is distributed on a global scale and infects travelers, children, and immunocompromised patients in endemic countries; in industrialized countries, people of any age are susceptible. Transmission is generally food-borne and associated with contaminated fresh produce. Usually, infection is self-limiting with the major symptom of acute diarrhea; however, in a small number of immunocompromised patients, chronic diarrhea can occur with extra-intestinal organ colonization [17].

Microsporidia are a diverse group of pathogens that infect a wide range of animals and contain 14 species that can infect humans [18]. Microsporidia infections in humans occur globally, and prevalence rates of between 0–50% have been recorded depending on the diagnostic technique used, geographical region, and the population studied [19]. The most well-known are *Enterocytozoon bieneusi* and *Encephalitozoon intestinalis,* which infect a number of domestic, farm, and wild animals, leading to the theory that infection is a zoonotic disease. Microsporidia have been detected in fresh vegetables and fruit and thus may also be transferred to the human population as a food-borne disease [20]. Microsporidian infection predominately occurs in HIV-infected or immunocompromised individuals with the main symptoms of weight loss, persistent diarrhea, and abdominal pain. However, in some patients, dissemination to the hepatobiliary, pulmonary, and other organ systems occurs, resulting in life-threatening conditions [19]. Table 1 shows a summary of the symptoms and number of infections occurring annually as a result of infection with *G. intestinalis*, *Cryptosporidium* spp., *E. histolytica*, and *C. cayetanensis.*

Traditionally, the diagnosis of many gastrointestinal parasites has been based on the detection of trophozoites or cysts by microscopical examination of stool samples. However, the sensitivity of microscopy can be low (46%) because of the sporadic shedding of cysts into stool. To improve the sensitivity of microscopy, it is recommended that three consecutive samples be taken, which results in an improvement of sensitivity up to 94% [21]. However, microscopy is time-consuming, labor-intensive, and requires highly trained staff. In order to increase the sensitivity and make diagnosis simpler, immunological assays have been developed for the detection of enteric protozoa. Such tests are easier to perform and useful for the rapid investigation of a large number of stool specimens [22]. Combination tests for the detection of *Giardia intestinalis*, *Cryptosporidium* spp., and *E. histolytica* in clinical samples, such as the Tri-Combo parasite screen test (TechLab, Blacksburg, VA, USA) and the Triage parasite panel (Biosite, San Diego, CA, USA), have been developed requiring minimal laboratory experience and reagents simplifying diagnosis further [23].

A wide range of PCR and RT-PCR-based techniques has been developed for the detection of intestinal parasites in clinical samples [24]. The advantage of these methods is that, in general, they tend to be more sensitive and specific compared to traditional microscopy and immunoassays [25]. Many commercially available RT-PCR tests exist that commonly detect the presence of *Giardia intestinalis*, *Cryptosporidium* spp., and *E. histolytica* directly from stool samples [26]. Basmaciyan et al. compared four commercially available tests, finding the assays were an alternative method compared to traditional techniques for the detection of the three protozoan targets in stool samples [27]. However, the authors noted differences in the sensitivity and specificity of each assay for the detection of *Giardia intestinalis* and *Cryptosporidium* spp., indicating that care should be taken when choosing an assay [27].

Table 2 compares the test menu of ten commercially available molecular assays that detect intestinal protozoan in clinical samples. All tests detect the presence of *G. intestinalis*, *Cryptosporidium* spp., and *E. histolytica*. Certest, SeeGene, Ausdiagnostic, and Genetic Signatures assays additionally detect *D. fragilis* and *B. hominis* with the Biofire, SeeGene, Ausdiagnostics, and Genetic Signatures assays detecting *C. cayetanensis*. The only assay capable of detecting all, including microsporidian targets, is the Genetic Signatures EP005 assay.

We previously developed a novel RT-PCR technology based on the bisulphite deamination of cytosine to thymine via a uracil intermediate, which simplifies conventional nucleic acid sequences containing adenine, thymine, guanine, and cytosine to a 3base^TM^ form consisting of just adenine, thymine, and guanine. The deamination reaction of cytosine to uracil was first described in 1970 by Hayatsu [28,29] and has been studied in detail since. The first step of the reaction involves the sulphonation of cytosine to cytosine sulphonate followed by deamination of cytosine to a uracil sulphonate intermediate and subsequently the removal of the sulphate adduct to uracil, traditionally by the use of strong alkali (Figure 1). 

After conversion, sequences become more similar at the nucleic acid level, allowing the detection of, for example, viruses from families containing large numbers of individual pathogens such as human papillomavirus [30], Norovirus GI and GII [31], and flavivirus [32]. In addition, the conversion of cytosine bases to thymine within target sequences contributes to improved PCR efficiency for organisms with high G-C content, such as *G. intestinalis*, where inefficiencies with amplification have been reported using conventional assays [33]. By way of demonstration, Table 3 shows the reduction in G-C content for *Entamoeba* spp., *G. intestinalis, D. fragilis, Cryptosporidium,* and *B. hominis* in an earlier version of our enteric protozoan detection kit, using a universal primer set and specific probes to differentiate the individual pathogens, except *Entamoeba*, where the Entamoeba complex was detected (*E. histolytica*, *E. dispar*, and *E. moshkovski*).

Using this 3base method, we developed a new mRT-PCR assay (*EasyScreen™* Gastrointestinal Parasite Detection Kit, EP005) capable of detecting the presence of eight of the most important protozoan parasites infecting the gastrointestinal tract of humans. All assays target the 18S/small subunit ribosomal RNA gene transcripts.

## 2. Results

Table 3 shows that using the 3base™ approach greatly reduces the G-C content of the primers and probe sequences. Before 3base™ conversion, the melting temperature of the primers varied by 11.4 °C, whereas after, there was only a 1.2 °C difference. Similarly, the melting temperature of the probes ranged from as low as 51.3 °C to a high of 78.8 °C, with the temperature difference reduced to 49–57.9 °C after treatment. Such large variations in temperature would lead to inefficient amplification of the targets as well as suboptimal probe binding. Therefore, the use of 3base™ can be advantageous when performing mRT-PCR where target sequences have a wide range of G-C content, thus normalizing PCR cycling conditions.

### 2.1. Assay Sensitivity

Table 4 shows the strains used to assess the lower limit of detection (LLOD) of the assays. All targets were diluted in a negative stool background, and then serial dilutions were extracted and purified prior to PCR amplification.

Table 5 shows that the LLOD of the assay components ranged from 4 organisms/mL for *Enterocytozoon bieneusi* to 5700 genome equivalents (GE)/mL for *Giardia intestinalis*.

To ensure the assay was capable of detecting a wide range of strains of each target organism, a number of cultures, clinical isolates, and DNA were obtained and tested at 3× the LLOD (See Table 6). All strains and cultured samples were successfully detected using the assay.

### 2.2. Assay specificity

Either cultured organisms or purified nucleic acids from the organisms listed in Table 7 were diluted to approximately 10^6^ CFU/mL or 10^9^ copies/mL, respectively, using a negative stool matrix. All samples were tested in triplicate to determine potential cross-reactivity with no amplification detected from EP005 using any of the non-target organisms.

### 2.3. Microbial Interference Studies

Each target organism was tested at 2× LLOD in a negative stool matrix containing either cultured organisms at 106 CFU/mL or purified nucleic acids at 109 copies/mL using the strains shown in Table 8. Each sample was then tested in triplicate to determine potential competitive inhibition with high concentrations of non-target organisms. All targets from EP005 were amplified at 2× LLOD.

### 2.4. Quality Assurance Panels

The assay was further validated with a quality assurance panel obtained from QCMD (Quality Controls for Molecular Diagnostics, Glasgow, Scotland) consisting of nine blinded samples. QCMD panels consist of whole inactivated organisms in a synthetic stool background. The panels were sent to participating laboratories to be coded, then samples were tested, and the results were returned before the panel content was released. As can be seen from Table 9, the results obtained were in agreement with the expected results.

### 2.5. Clinical Studies

To assess clinical sensitivity, 380 samples were tested with both the BD MAX™ Enteric Parasite Panel and the *EasyScreen*™ Gastrointestinal Parasite Detection Kit, EP005 (Table 10). The BD MAX™ Enteric Parasite Panel detects *G. intestinalis*, *Cryptosporidium* spp., and *E. histolytica*. In order to confirm the presence of pathogens not covered in the BD MAX™ panel, individual conventional 4base primers sets were designed for *D. fragilis*, *C. cayetanensis*, *B. hominis*, *E. bieneusi*, and *E. intestinalis*. Samples positive for these targets were re-amplified with these assays and sequenced to confirm the identity of the pathogen detected.

Table 11 shows the sensitivity, specificity, positive predictive value, and negative predictive value of the EP005 assay compared to the gold standard BDMax™ assay. As can be seen from the data, the EP005 assay showed a high level of sensitivity and specificity compared to the comparator assay.

*Cryptosporidium* was detected in 10 samples using the EP005 assay (2.6%), and 9 of these were also positive using the BD MAX™ assay. Twenty samples tested positive for *G. intestinalis* using EP005 (5.3%), with an extra positive sample found when using the BD MAX™ assay. Eight samples tested positive for *E. histolytica* using both methods (2.1%).

Thirty-seven samples (9.7%) were positive for *D. fragilis,* and sixteen of these samples were selected for re-amplification and sequencing, with fifteen samples tested confirmed as *D. fragilis. B. hominis* was detected in a total of 24/380 (6.3%) samples, and again, sixteen of the positive samples were analyzed by sequencing, confirming eleven as *B. hominis*. The increase in positive samples using the EP005 assay was due to the improved sensitivity of EP005 compared to the NAAT assays.

*C. cayetanensis* was detected in eleven retrospective samples, and again, all samples were confirmed to contain *C. cayetanensis* on sequencing. In a retrospective study, twenty-one samples that had previously tested positive for *E. bieneusi* using an alternative assay were tested, and all samples produced positive amplification curves for *E. bieneusi*. Nineteen of twenty-one of these samples tested positive on sequencing; however, two samples were negative as a result of the reduced sensitivity of the NAAT primers sets. Five samples were positive for *E. intestinalis* using both the 3base™ assay and PCR followed by sequencing. Finally, 3% of samples contained mixed infection with two or more parasites detected. The Ct values of all positive samples detected in the study are shown in Appendix A.

## 3. Discussion

The three most commonly occurring protozoan parasites that infect humans are *G. intestinalis*, *Cryptosporidium* spp., and *E. histolytica*. A number of other species, including *C. cayetanensis, E. bieneusi,* and *E. intestinalis,* are well-established disease-causing protozoa that infect the gut. Both *D. fragilis* and *B. hominis* have been implicated in the pathogenesis of gastrointestinal disease for many years and are commonly found in the intestinal tract.

Traditional methods used for the detection of these parasites have been microscopy and immunoassays. However, microscopy is time-consuming, and the results require highly trained technicians. Immunoassays suffer from reduced sensitivity and specificity and can have trouble differentiating pathogenic *E. histolytica* from the non-pathogenic *E. dispar* and *E. moshkovskii* compared to PCR-based methodologies. Both conventional and real-time PCR have become well-established techniques for the detection of protozoan parasites. In general, PCR methods are easy to perform, more sensitive, and have the added advantage that they can be used to detect multiple pathogens in the same sample.

Most commercially available PCR assays target *G. intestinalis*, *Cryptosporidium* spp., and *E. histolytica,* with some additionally targeting *D. fragilis*, *B. hominis*, or *C. cayetanensis.* Protozoan infections can be successfully treated using a number of drugs, including metronidazole, tinidazole, and nitazoxanide; thus, it is vital to have rapid and sensitive assays that can detect the parasite responsible for disease and to administer appropriate anti-protozoan drugs if required. To date, no commercial assay is available that detects all these parasites and the microsporidian targets. We developed a novel assay that targets all eight human protozoan parasites. The method has the advantage that the primers and probes used in the assays have a more similar melting temperature, which improves the efficiency of mRT-PCR.

The analytical sensitivity of the method was shown to be: 4 organisms/mL for *E. bieneusi*, 25 GE/mL for *Cyclospora cayetanensis*, 100 organisms/mL *B. hominis*, 125 org/mL for *D. fragilis,* 1600 org/mL for *Cryptosporidium*, and 1200–2500, 3000, and 5700 organisms or GE/mL for *E. intestinalis, E. histolytica*, and *Giardia intestinalis*, respectively. Given that the assays target the 18S/small subunit ribosomal RNA gene transcripts, the differing sensitivities may partially reflect the different copy numbers and expression levels of the 18S rRNA genes in different species, cultures, and clinical isolates. Testing a range of different isolates and cultures confirmed that the assay was able to detect all strains that were analyzed. No cross-reactivity was seen using a wide range of organisms that would be commonly found in stool samples.

Previously, a number of studies have compared our *EasyScreen*^TM^ Enteric Detection Kit EP001 to traditional ova and parasite (O&P) examinations [31,34,35]. In a study analyzing 358 samples, the EP001 kit yielded 92–100% sensitivity (compared to 55–77% for traditional O&P) and 100% specificity (compared to 95–100% for traditional O&P) for the detection of *Cryptosporidium* spp., *G. intestinalis*, *Entamoeba* spp., *D. fragilis*, and *B. hominis* [35].

A clinical trial was conducted to assess the performance of the new EP005 assay consisting of 380 stool samples, and the results were compared to another commercially available assay, the BD MAX™ Enteric Parasite Panel, which detects the presence of *G. intestinalis*, *Cryptosporidium* spp., and *E. histolytica.* To confirm the presence of pathogens not included in the BD MAX™ system, alternative PCR assays were designed, and positive samples were re-amplified with these primers and then sequenced to determine the identity of the suspected pathogen. Overall, there was a high level of agreement of the assay with the BD MAX™ with 10 samples positive for *Cryptosporidium* spp. using the EP005 assay compared to the 9 for the BD MAX™; 20 samples were positive using the EP005 assay for *G. intestinalis* compared to the 21 for the BD MAX™, and 8 samples were positive for *E. histolytica* using both. The use of alternative PCR assays and sequencing for the detection of pathogens not targeted by the BD MAX™ system showed a high degree of concordance with the expected results. Interestingly, 3% of samples contained mixed infections, demonstrating the utility of the multiplex assay to detect co-infections.

mRT-PCR technologies will most likely never be a substitute for microscopy as they do not detect the presence of all enteric protozoa or parasites that can cause infection. The EP005 assay uses automated sample extraction technologies, simplifying the process of nucleic acid purification from stool samples and decreasing the involvement of highly trained employees, thus reducing hospital costs and streamlining laboratory workflows. The EP005 assay is a sensitive and specific assay for the detection of enteric protozoan in clinical samples and has the most comprehensive assay menu available. Results are generated within 4 h; thus, drugs can be prescribed in a timely manner if required, ultimately improving patient care.

## 4. Materials and Methods

### 4.1. EP005 PCR Assays

Real-time PCR assays were designed against 18S/small subunit ribosomal RNA consensus sequences obtained from public databases (Genbank, NCBI, Bethesda, MD, USA). Sequences were first 3base converted by changing all C bases to T in the sense strand. Some targets (*D. fragilis*, *Cryptosporidium* spp., *B. hominis*, and *G. intestinalis*) use common forward and reverse primers but specific fluorescent probes for each target. The other four targets use unique primers and probes. Sequences are commercial in confidence. To formulate EP005, targets were divided into three multiplex panels per Table 12 below. Each panel also includes a control; Panels A and C contain an extraction control (EC) pan-bacterial 16S ribosomal RNA target to ascertain sample quality (i.e., bacterial load), and panel B contains an internal positive control (IPC) to indicate any PCR inhibition, whereby amplification should occur within a defined cycle threshold (Ct) range in the absence of inhibition. Each probe fluoresces at a given wavelength, and the signals are measured and distinguished from each other by the real-time PCR platform. The real-time PCR software provided with each instrument interprets all data collection and provides the information for automated or manual result analysis.

### 4.2. Preparation of Negative Stool

Stool specimens known to be free of the eight EP005 targets, either by microscopy or using EP001, were processed using the SP008B sample-processing kit according to the instructions (Genetic Signatures, Sydney, Australia) on the GS1 automated sample-processing platform and amplified with the *EasyScreen*™ Gastrointestinal Parasite Detection kit (EP005) to confirm the absence of EP005 targets. Negative samples were then pooled, and 10 volumes of PBS were added per gram of stool and re-tested using EP005. The confirmed negative stool matrix was then stored at −80 °C until required.

### 4.3. LLOD and Inclusivity Studies

Target organisms were quantified using synthetic double-stranded DNA (gBlocks, IDT, Sydney, Australia), listed in Appendix A. gBlocks were serially diluted and compared to DNA extracted from the target organisms. A standard curve was generated using the Bio-Rad CFX Manager 3.1 Software, and this was used to calculate the copy number of each target. The quantified targets were then diluted into a negative stool background and extracted on the GS1 automated platform. The eluates were PCR-amplified using the EP005 assay (Genetic Signatures, Sydney, Australia). The acceptance criteria for LLOD were the concentration at which 95% or greater detection of the specified target was achieved.

The inclusivity of the *EasyScreen*™ Gastrointestinal Parasite Detection Kit (EP005) was also evaluated using a collection of 37 isolates (ATCC culture and quantified sequenced clinical specimens) representing the eight EP005 targets. Targets were tested at no higher than 3 times the LLOD in a negative stool matrix. Samples were extracted and tested in triplicate, with 100% detection for the specified target as the acceptance criteria.

### 4.4. Cross-Reactivity Studies

Cultured organisms or DNA samples were obtained from ATCC and diluted to 10^6^ CFU/mL or 10^9^ copies/mL, respectively, in a negative stool background. Samples were then tested in triplicate, according to Section 4.2.

### 4.5. Clinical Samples

In total, 380 clinical samples were obtained from local Sydney hospitals and universities, and these were processed according to Section 4.2. In addition, the samples were also processed using the BD MAX™ Enteric Parasite Panel according to the manufacturer’s instructions.

### 4.6. Confirmatory Nucleic Acid Amplification Techniques

Table 13 lists the sequences of the primers used to confirm the results generated using the EP005 system that was not covered by the BD MAX™ Enteric Parasite Panel. Primers were synthesized by IDT technologies (Sydney, Australia). PCR conditions were as follows: 42 °C 15 min, 95 °C 3 min; 95 °C 15 s 62–65 °C for 60 s repeated 50 cycles. The SsoAdvanced Universal SYBR Green Supermix (BioRad, Sydney, Australia) was used as a PCR mastermix. All positive samples were sent for sequencing using the same forward and reverse primers used in the RT-PCR reaction. The PCR product (~17 μL) and 100 μL of 3.2 μM of each primer were sent to AGRF Molecular Genetics (Sydney, Australia) for PCR cleanup and Sanger sequencing.

## Figures and Tables

**Figure 1 ijms-24-13387-f001:**
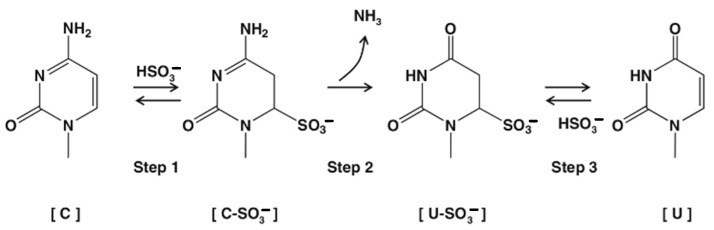
Bisulphite mediated conversion of cytosine to uracil.

**Table 1 ijms-24-13387-t001:** Symptoms and number of infections annually worldwide for *G. intestinalis*, *Cryptosporidium* spp., *E. histolytica*, and *C. cayetanensis*.

Organism	Symptoms	Infections
*G. intestinalis*	Acute, watery diarrhea. Chronic infections can result in weight loss and malabsorption and can cause long-term complications, including irritable bowel syndrome and chronic fatigue.	>200 million people annually.
*Cryptosporidium* spp.	Symptoms include abdominal pain, fever, vomiting, malabsorption, and self-limiting diarrhea.	>2.9 million cases of cryptosporidiosis in children aged <2 years old in sub-Saharan Africa alone annually.
*E. histolytica*	Asymptomatic to severe symptoms, including dysentery (bloody diarrhea) and extra-intestinal abscesses.	Amebiasis results in 100,000 deaths annually on a global basis.
*C. cayetanensis*	Watery diarrhea, weight loss, and cramping.	Outbreaks of cyclosporiasis occur regularly in the US, mostly linked to infected food or drink.

**Table 2 ijms-24-13387-t002:** Commercially available enteric protozoan assays.

	Manufacturer									
	Becton Dickinson (Franklin Lakes, NJ, USA)	Diagenode (Denville, NJ, USA)	R-Biopharm (Hessen, Germany)	Fast Track (Esch-sur-Alzette, Luxembourg)	Savyon (Ashdod, Israel)	Certest (Zaragoza, Spain)	Biofire (Salt Lake City, UT, USA)	SeeGene (Seoul, Republic of Korea)	AusDiagnostics (Mascot, NSW, Australia)	Genetic Signatures (Newtown, NSW, Australia)
Target	BD MAX™	Gastro panel	RIDAGENE	FTD Stool	NanoCHIP	VIASURE	Filmarray	ALLplex	Parasite 8	EP005
*G. intestinalis*	Yes	Yes	Yes	Yes	Yes	Yes	Yes	Yes	Yes	Yes
*Cryptosporidium* spp.	Yes	Yes	Yes	Yes	Yes	Yes	Yes	Yes	Yes	Yes
*E. histolytica* *	Yes	Yes	Yes	Yes	Yes	Yes	Yes	Yes	Yes	Yes
*D. fragilis*	No	No	No	No	No	Yes	No	Yes	Yes	Yes
*B. hominis*	No	No	No	No	No	Yes	No	Yes	Yes	Yes
*C. cayetanensis*	No	No	No	No	No	No	Yes	Yes	Yes	Yes
*E. bieneusi*	No	No	No	No	No	No	No	No	No	Yes
*E. intestinalis*	No	No	No	No	No	No	No	No	No	Yes

* All assays listed are specific for *E. histolytica* and do not detect *E. dispar* or *E. moshkovski.*

**Table 3 ijms-24-13387-t003:** Examples of the reduction in melting temperature that can be achieved in mRT-PCR using 3base™ technology.

	4Base Sequence		3Base™ Sequence	
Target	Sequence	Tm °C	Sequence	Tm °C
Primer-1	GTACACACCGCCCGTCGCTCCTACC	66.6	GTATATATTGTTTGTTGTTTTTATT	51
Primer-2	GAGGAAGGTGAAGTCGTAACAAG	55.2	GAGGAAGGTGAAGTTGTAATAAG	49.7
*Entamoeba* spp. Probe	TGAATAAAGAGGTGAAATTCTAGG	51.3	TGAATAAGAGGTGAAATTTTAGG	49
*G. intestinalis* Probe	GCGCGAAGGGCCGCGAGCCCCCGCGC	78.8	GTGTGAAGGGTTGTGAGTTTTTGTGT	57.9
*D. fragilis* Probe	TTTAAATGAAAAGGTGATTAAATCACG	51.2	TTTAAATGAAAAGGTGATTAAATTATG	47.7
*Cryptosporidium* spp. Probe	GACCATACTTTGTAGCAATACATGTAAGGA	56.6	GATTATATTTTGTAGTAATATATGTAAGGA	48.1
*B. hominis* Probe	TATTGAAAGAAGTTGTGTAAATCTTACCATT	53.9	TATTGAAAGAAGTTGTGTAAATTTTATTATT	50

**Table 4 ijms-24-13387-t004:** Organisms used to assess the LLOD of each target.

Target Organism; Strain (Accession Number)	Target Type (ATCC Cat. Number)
*Blastocystis hominis* NANDII	Culture (50177)
*Blastocystis hominis* NTY	Culture (50610)
*Cryptosporidium parvum*; (* CP082114.1)	Clinical stool
*Cryptosporidium hominis*; (* XM_661199.1)	Clinical stool
*Cyclospora cayetanensis*; (* AF111183.1)	Clinical stool
*Dientamoeba fragilis*; (* JQ677148.1)	Culture
*Dientamoeba fragilis*; (* JQ677148.1)	Clinical stool
*Entamoeba histolytica* HU-21:AMC	Culture (30457)
*Enterocytozoon bieneusi*; (* AF023245.1)	Clinical stool
*Enterocytozoon bieneusi*; MWC_m4 (MG976813.1)	Clinical stool
*Encephalitozoon intestinalis*	Culture (50506)
*Giardia lamblia,* Human isolate—H3 (AF023245.1)	Quantified non-viable oocysts
*Giardia intestinalis*; BE-1	Culture PRA-42

* The accession number was the best match for sequencing of the PCR amplicons.

**Table 5 ijms-24-13387-t005:** LLOD of each target in the EP005 panels.

	LLOD	Positive Replicates		Ct Value Average	
Target	Concentration	Isolate 1	Isolate 2	Isolate 1	Isolate 2
*Dientamoeba fragilis*	125 org/mL	20/20	20/20	32.45	30.66
*Cryptosporidium parvum*	1600 org/mL	19/20	20/20	32.54	31.95
*Cryptosporidium hominis*	1600 org/mL	20/20	20/20	31.81	30.34
*Cyclospora cayetanensis*	25 GE/mL	20/20	20/20	33.11	31.76
*Blastocystis hominis*	100 org/mL	20/20	19/20	31.96	33.56
*Giardia intestinalis*	5700 GE/mL	20/20	19/20	32.23	32.63
*Entamoeba histolytica*	3000 org/mL	20/20	20/20	33.76	32.81
*Enterocytozoon bieneusi*	4 org/mL	20/20	20/20	34.53	32.79
*Encephalitozoon intestinalis*	1200–2500 org/mL	20/20 *	20/20 #	33.04	29.84

* Isolate 1 was tested at 1200 organisms/mL, and # Isolate 2 was tested at 2500 organisms/mL.

**Table 6 ijms-24-13387-t006:** Assay inclusivity using clinical samples and cultured isolates.

Organism; Strain (Accession Number)	Target Type *	Organism; Strain (Accession Number)	Target Type *
*Blastocystis hominis*; BT1	Culture (50608)	*Dientamoeba fragilis*	Clinical
*Blastocystis hominis*; NMH	Culture (50754)	*Dientamoeba fragilis*	Clinical
*Blastocystis hominis*; DL	Culture (50626)	*Dientamoeba fragilis*	Clinical
*Cryptosporidium parvum*	Clinical	*Encephalitozoon intestinalis*; CDC:V297 (50651)	Culture (50651)
*Cryptosporidium parvum*	Clinical	*Encephalitozoon intestinalis*; nasal isolate (50507)	Culture (50507)
*Cryptosporidium parvum*	Clinical	*Encephalitozoon intestinalis*; CDC:V307 (50603)	Culture (50603)
*Cryptosporidium parvum*	Clinical	*Encephalitozoon intestinalis*; CDC:V308 (50790)	Culture (50790)
*Cryptosporidium parvum*	Clinical	*Enterocytozoon bieneusi*	DNA
*Cryptosporidium hominis*	Clinical	*Entamoeba histolytica*; HB-301:NIH (30190)	Culture (30190)
*Cryptosporidium hominis*	Clinical	*Entamoeba histolytica*; HK-9 Clone 6 (50544)	Culture (50544)
*Cryptosporidium hominis*	Clinical	*Entamoeba histolytica*; HB-301:NIH CL-1-3 (50547)	Culture (50547)
*Cryptosporidium hominis*	Clinical	*Giardia intestinalis*; WB (30957)	Culture (30957)
*Cryptosporidium hominis*	Clinical	*Giardia intestinalis*; WB clone C6 (50803)	Culture (50803)
*Dientamoeba fragilis*	Clinical	*Giardia intestinalis*; GS clone H7 (50581)	Culture (50581)
*Cyclospora cayetanensis* (KX618190)	Clinical	*Giardia intestinalis*; Portland-1 (30888)	Culture (30888)
*Cyclospora cayetanensis* (MN316534)	Clinical	*Giardia intestinalis*; (Lambl) Alexeieff Strain PR-15	Culture
*Cyclospora cayetanensis* (MN316534)	Clinical	*Giardia lamblia*; CM (PRA242)	Culture (PRA242)
*Cyclospora cayetanensis* (MN316535)	Clinical		
*Dientamoeba fragilis*	Clinical		

* Where applicable, the ATCC catalogue numbers have been included in brackets. For clinical samples, the accession number for the best match on sequencing of the PCR amplicon is listed.

**Table 7 ijms-24-13387-t007:** Species used to assess assay specificity.

Cross-Reactivity Species Tested			
*Atopobium vaginae* *	*Cedecea davisae* *	Human adenovirus 5	*Proteus vulgaris*
*Abiotrophia defective* *	*Chlamydia trachomatis*	*Klebsiella oxytoca*	*Providencia stuartii*
*Acinetobacter baumannii*	*Chryseobacterium gleum* *	*Lactobacillus acidophilus*	Rotavirus A
Adenovirus	*Citrobacter freundii*	*Lactococcus lactis*	*Saccharomyces cerevisiae* *
*Aeromonas hydrophila*	*Clostridium perfringens*	*Leminorella grimontii* *	*Salmonella enterica*
*Akkermansia muciniphila*	*Corynebacterium glutamicum* *	*Listeria monocytogenes*	Sapovirus
*Alcaligenes faecalis*	*Cronobacter sakazakii*	*Mycobacterium abscessus*	*Serratia marcescens*
*Anaerococcus tetradius* *	*Desulfovibrio pigerl* *	*Mycobacterium avium*	*Shigella flexneri*
*Arcobacter butzleri* *	*Edwardsiella tarda* *	*Mycobacterium* spp.	*Shigella sonnei*
Astrovirus	*Eggerthella lenta*	*Mycoplasma fermentans*	*Staphylococcus aureus*
*Atopogium vaginae* *	*Enterococcus faecalis*	*Mycoplasma hominis*	*Staphylococcus hominis*
*Bacillus cereus*	*Enterococcus faecium*	*Mycoplasma salivarium*	*Stenotrophomonas maltophila*
*Bacteroides fragilis*	Enterovirus	*Neisseria flava*	*Streptococcus mitis*
*Bifidobacterium adolescentis*	*Escherichia coli*	Norovirus GI	*Streptococcus pyogenes*
Bocavirus	*Eubacterium rectale* *	Norovirus GII	*Streptococcus salivarius*
*Campylobacter coli*	*Faecalibacterium prausnitzii* *	*Peptoniphilus asaccharolyticus* *	*Streptococcus sanguinis*
*Campylobacter hominis*	*Fusobacterium varium* *	*Peptostreptococcus anaerobius*	*Streptococcus thermophilus*
*Campylobacter jejuni*	*Gardnerella vaginalis*	*Porphyromonas asaccharolytica* *	*Veillonella parvula*
*Campylobacter lari*	*Gemella morbillorum* *	*Porphyromonas levii* *	*Yersinia pseudotuberculosis*
*Candida albicans*	*Hafnia alvei*	*Prevotella melaninogenica*	
*Capnocytophaga gingivalis*	*Helicobacter pylori*	*Proteus mirabilis*	

* Cultured isolates.

**Table 8 ijms-24-13387-t008:** Organisms used in the competitive interference study.

Target Organism	Target Type (ATCC Cat. Number)
*Pseudomonas aeruginosa*	ATCC 47085DQ
*Candida albicans*	ATCC MYA-2876D-5
*Clostridioides perfringens*	ATCC 13124DQ
non-pathogenic *Escherichia coli*	ATCC 25922DQ
*Enterococcus faecalis*	ATCC 700802DQ
*Bacteroides fragilis*	ATCC 25285D-5
*Klebsiella pneumoniae*	ATCC 13883DQ
*Saccharomyces cerevisiae* *	ATCC MYA-796

* Cultured isolate.

**Table 9 ijms-24-13387-t009:** Results generated with the QCMD quality assurance panel.

Sample	QCMD Result	EP005 Result
EP19S-01	*D. fragilis/B. hominis*	*D. fragilis/B. hominis*
EP19S-02	*E. histolytica*	*E. histolytica*
EP19S-03	Negative	Negative
EP19S-04	*E. histolytica*	*E. histolytica*
EP19S-05	*G. intestinalis*	*G. intestinalis*
EP19S-06	*Cryptosporidium* spp.	*Cryptosporidium* spp.
EP19S-07	*G. intestinalis*	*G. intestinalis*
EP19S-08	*Cryptosporidium* spp.	*Cryptosporidium* spp.
EP19S-09	*G. intestinalis*	*G. intestinalis*

**Table 10 ijms-24-13387-t010:** Results generated comparing the assay to an alternative commercially available test and PCR and sequencing with other nucleic acid amplification techniques (NAAT).

Target	Specimen Type	Samples Tested	EP005 Positive	Confirmatory Method	Positive with Confirmatory Method
*Cryptosporidium* spp.	Prospective	380	10	BD MAX™	9
*G. intestinalis*	Prospective	380	20	BD MAX™	21
*E. histolytica*	Prospective	380	8	BD MAX™	8
*B. hominis*	Prospective	380	24	NAAT	11/16 *
*D. fragilis*	Prospective	380	37	NAAT	15/16 *
*C. cayetanensis*	Retrospective	11	11	NAAT	11
*E. bieneusi*	Retrospective	21	21	NAAT	19 **
*E. intestinalis*	Contrived/Retrospective	5	5	NAAT	5

* 16 samples of each were selected for sequencing with five *B. hominis* and one *D. fragilis* samples negative by NAAT due to the improved sensitivity of the 3base™ assay. ** 2 samples were negative by NAAT due to the improved sensitivity of the 3base™ assay.

**Table 11 ijms-24-13387-t011:** Specificity, sensitivity, PPV, and NPV of the EP005 assay compared to the BDMax™ assay.

Target	Sensitivity (95% CI)	Specificity (95% CI)	PPV (95% CI)	NPV (95% CI)
*Cryptosporidium* spp.	100% (66.37–100)	99.7% (98.51–99.99)	90% (55.97–98.46)	100%
*G. intestinalis*	95.45% (77.16–99.88)	100% (98.98–100)	100%	99.74% (98.14–99.96)
*E. histolytica*	100% (63.06–100)	100% (99.01–100)	100%	100%

**Table 12 ijms-24-13387-t012:** Panel and target composition of the EP005 kit.

Channel/Representative Fluorophore	Panel A	Panel B	Panel C
Fam	*D. fragilis*	*B. hominis*	*E. bieneusi*
Hex/Vic	EC	IPC	EC
Texas Red/Rox	*C. cayetanensis*	*E. histolytica*	*E. intestinalis*
Cy5	*Cryptosporidium*. spp.	*G. intestinalis*	Not used

**Table 13 ijms-24-13387-t013:** Shows the sequences of the alternative NAAT assays used as confirmatory assays in the clinical study.

Target	Forward Primer	Reverse Primer	Gene	Anneal Temp
*D. fragilis* Region 1	AATACCTTTTAATAGGTAATCCAATCGA	GCCCTCTGCTAGGTTACAATATAC	18S rRNA	64 °C
*D. fragilis* Region 2	TTTAATGACTGATCAGGCTATAGG	CATCACGGACCTGTTATTGCTACCA	18S rRNA	64 °C
*C. cayetanensis* Region 1	CGCATTTGGCTTTAGCCGGCGATA	TTACTCTGGAAGGATTTTAAATTCCT	18S rRNA	62 °C
*C. cayetanensis* Region 2	CACGCTCTACCAATATTCGTTATCA	GGATCGTGTTGGCTAGGTGTACTAA	Mitochondria	65 °C
*B. hominis* Region 1	AGTAGTCATACGCTCGTCTCAAA	TCTTCGTTACCCGTTACTGC	SSU RNA	65 °C
*B. hominis* Region 2	CAATTGGAGGGCAAGTCTGGTGC	CACCTCTAACTATTGAATATGAATACC	SSU RNA	64 °C
*E. bieneusi* Region 1	TAGCGGAACGGATAGGGAGTGTAGT	CTTGCGAGCGTACTATCCCCAGAG	SSU RNA	64 °C
*E. bieneusi* Region 2	CATTCGTTGATCGAATACGTGAGAAT	GTTACTAGGAATTCCTTATTCACTACG	SSU RNA	64 °C
*E. intestinalis* Region 1	TCACGGCATCCATTTCAAACGG	GATGAAGGACGAAGGCTAGAGGA	16S rRNA	64 °C
*E. intestinalis* Region 2	GTCCAAGAGCACAGCCTTCGCTTC	GGGATCGGGGTTTGATTCCGGA	16S rRNA	64 °C

## Data Availability

The data presented in this study are available on request from the corresponding author. The data are not publicly available due to commercial in confidence.

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
