# Peer review of "The Application of 3base™ Technology to Diagnose Eight of the Most Clinically Important Gastrointestinal Protozoan Infections"

_ijms, 2023, doi:10.3390/ijms241713387_

Round 1

Reviewer 1 Report

Dear authors. 

The manuscript is significant in detecting species associated with infectious intestinal parasites.

Lanes below, you will find some observations:

Lane 18. We usually do not include references in the abstract. Furthermore, all abstracts should describe what the authors have studied in the research, what they found, and what the authors argue in the manuscript.

The actual version is a partial introduction not well conducted.

Lane 54, references are missing.

Lane 99, reference is missing for the Tri-Combo parasite screening test.

Lane 103. Reference is missing

Lane 105. Reference is missing

Lane 107. Reference is missing.

Lane 108. Acceptable performance meaning?

Lane 111. Significant difference based on which statistical method? Which kind of samples? Which sample sizes were analyzed?

lane 159: There is a typo on the subtitle “sesnsitivity”

Tables 4 and 6: Accession numbers for all strains are not indicated. Information is missing.

Lane 181: Typo on the subtitle: “Spcificity”

Table 6. What is the number of samples evaluated?

Table 7. Where is the information, regardless of the specificity?

Quality control:

Number of samples evaluated? Is the number of samples enough to mention a 100% agreement?

Which DNA concentration was used in the mRT-PCRs assays for quality control?

Assay specificity: Which samples were diluted 106 and which ones in the interval 106-109?

Please show the threshold cycle value (Ct) for every sample evaluated by mRT-PCR and Ct for each assay (specificity, sensibility).

Please also include the raw data (in a supplementary file) to see the amplification data for all samples evaluated (including the Ct).

Lanes 212-227, please generate a table of sensitivity and specificity obtained for mRT-PCR compared with BD MAXTM assay.

The authors mentioned that the samples were sequenced to confirm the specificity. Please include the accession number ID (NCBI id) for all fragments sequenced in the manuscript.

Although some of the requested information is described somewhere in the manuscript, it is not mentioned in the description of the results and discussion section, which needs to be clarified.

The manuscript has several typos. Some of them are mentioned in the comments section.

Reviewer 2 Report

This research establishes the convenient detection of several intestinal protozoa parasite systems with RT-PCR. They are important and effective to construct an epidemiological intestinal protozoa parasite database.

I have some questions and comments as follows.

(introduction)

It would be better for the authors to mention the point of improvement of EP005 compared with EP001 and clarify the point discussed in this manuscript in the Introduction session.

(Table1)

About the item of “Transmission”, the transmission routes of Giardia, Cryptosporidium, Entamoeba histolytica, and Cyclospora are similar. So, it would be better to unify the terms used. As a special note, Giardia and Cryptosporidium are zoonoses. And E.histolytica is well known as Sexually Transmitted Disease (STD).

(Table3)

There is no probe information about Cyclospora cayetanensis, Enterocytozoon

bieneusi and Enterocytozoon intestinalis.

There is no description of the target gene locus.

Can Giardia probe detect all assemblage (A-H) or only

human assemblage (A and B)?

Can Entamoeba spp. probe distinguishes E.histolytica and other Entamoeba spp. (such as E.dispar, E.moshkovskii, and E.coli, etc.), if they are mixed infection?

(Assay sensitivity)

Are these measured sensitivities guaranteed even though mixed infection?

(Clinical studies)

To assess the clinical sensitivity of EP005, authors screened 380 clinical samples with EP005 and BDmax (as the gold standard) for Cryptosporidium spp, Giardia intestinalis, and Entamoeba historytica. However, only a limited number of samples were tested for B.hominis, D.fragilis, C.cayetanensis, E.bieneusi, and E.intestinalis (EP005 and NAAT). So, it looks difficult to assess sensitivity for these 5 species. If possible, it would be better to screen all samples with NAAT.
